# Modulating PAK1: Accessory Proteins as Promising Therapeutic Targets

**DOI:** 10.3390/biom15020242

**Published:** 2025-02-07

**Authors:** Amin Mirzaiebadizi, Rana Shafabakhsh, Mohammad Reza Ahmadian

**Affiliations:** 1Institute of Biochemistry and Molecular Biology II, Medical Faculty, Heinrich Heine University Düsseldorf, 40225 Düsseldorf, Germany; amin.mirzaiebadizi@hhu.de; 2Institute for Experimental Molecular Imaging, RWTH Aachen University Hospital, 52074 Aachen, Germany; rshafabakhsh@ukaachen.de

**Keywords:** PAK1, p21-activated kinase, accessory proteins, scaffold proteins, adaptor proteins, α/βPIX, CKIP1, GIT1, GRB2, NCK1, paxillin, RIT1, accessory inhibitors, merlin, nischarin, PAK1P1

## Abstract

The p21-activated kinase (PAK1), a serine/threonine protein kinase, is critical in regulating various cellular processes, including muscle contraction, neutrophil chemotaxis, neuronal polarization, and endothelial barrier function. Aberrant PAK1 activity has been implicated in the progression of several human diseases, including cancer, heart disease, and neurological disorders. Increased PAK1 expression is often associated with poor clinical prognosis, invasive tumor characteristics, and therapeutic resistance. Despite its importance, the cellular mechanisms that modulate PAK1 function remain poorly understood. Accessory proteins, essential for the precise assembly and temporal regulation of signaling pathways, offer unique advantages as therapeutic targets. Unlike core signaling components, these modulators can attenuate aberrant signaling without completely abolishing it, potentially restoring signaling to physiological levels. This review highlights PAK1 accessory proteins as promising and novel therapeutic targets, opening new horizons for disease treatment.

## 1. General Introduction

PAK1 is a key member of the PAK family of serine/threonine protein kinases, which consists of type I (PAK1–3) and type II (PAK4–6) PAKs, each with distinct regulatory mechanisms. As an effector of the RHO GTPases RAC1 and CDC42 [1], PAK1 plays a pivotal role in cellular processes related to cytoskeletal remodeling, including cell contraction, adhesion, motility, and polarity [2,3,4,5,6]. Once activated, PAK1 controls multiple proliferation and survival pathways, including MAPK, AKT, Wnt/β-catenin, and NF-κB [4]. Importantly, the PAK1 function is tightly regulated by its precise spatiotemporal localization at the plasma membrane, which depends on its recruitment and activation [7].

Amplification of the PAK1 gene and high levels of PAK1 protein are often associated with poor clinical prognosis, invasive tumor characteristics, and resistance to therapy [4,8,9]. Mutations in upstream GTPases, such as RAC1 [10], RAC2 [11], and RAS [12], can lead to the hyperactivation of PAK1, linking oncogenic signaling to phenotypic changes in cancer cells. In addition to cancer progression, PAK1 functions have been implicated in aging, neurodevelopmental disorders, liver disease, immune system abnormalities, and cardioprotection [13,14,15,16,17,18]. Given these diverse roles, exploring novel strategies to inhibit PAK1 is critical, particularly in light of the genetic evidence and the observed acute toxicities associated with various types of PAK1 inhibitors. The next sections will discuss several aspects of PAK1 regulation, modulation, and functions.

## 2. Mechanism of PAK1 Activation

PAK1 is activated by the CDC42 and RAC groups of small GTPases through several upstream signals, including estrogen, insulin, EGF, PDGF, IGF-1, and thrombin [19,20,21]. PAK1 (60.6 kDa; 545 amino acids or aa) consists of two distinct regions (Figure 1A) [22]. The conserved N-terminal non-catalytic region contains two overlapping domains, namely the p21-binding domain (PBD; aa 67–113) and the autoinhibitory domain (AID; aa 83–149). It also contains three conserved proline-rich regions (aa 12–18, 40–45, and 186–203) that serve as binding sites for several accessory proteins. An integral part of the PBD is the CDC42/RAC interactive binding (CRIB; aa 75–90) motif, where the binding of CDC42 or RAC1 initiates PAK1 activation [23,24]. This activation likely requires the association of PAK1 with the acidic phospholipid of the plasma membrane via a basic region (BR; aa 48–72) [25]. The C-terminal region of PAK1 contains a conserved Ser/Thr kinase domain (aa 250–520) with a single phosphorylation site (Thr-423) that is phosphorylated upon PAK1 activation [26].

Inactive PAK1 has been proposed to exist in an asymmetric trans-autoinhibited dimeric form [27,28,29], in which the AID of one PAK1 molecule contacts the kinase domain of the other, stabilizing the catalytic site in an autoinhibited state. The recruitment and activation of PAK1 at the plasma membrane require interactions at two sites, namely PBD binding with membrane-associated CDC42 or RAC1 and the BR binding with acidic phospholipids, particularly phosphatidylinositol-4,5-bisphosphate [25]. These events lead to the release of AID from the kinase domain, dimer/monomer equilibrium, and subsequent phosphorylation at Thr423 [30]. Once activated, PAK1 phosphorylates its substrates in a context-dependent manner that is tightly modulated by accessory proteins (Figure 1B).

Phosphorylation plays a critical role in stabilizing the active conformation of PAK1 and modulating its downstream signaling functions. In addition to phosphorylation sites within its activation loop, multiple phosphorylation events occur in the N-terminal non-catalytic region, affecting PAK1 activity and protein interactions [31]. PAK1 phosphorylation sites can be divided into three groups, namely (i) autophosphorylation sites, where PAK1 phosphorylates itself to regulate activation and substrate interactions (Ser21, Ser57, Ser144, Ser149, Ser199, and Ser204); (ii) phosphorylation sites modified by other kinases, including Tyr131, phosphorylated by SRC family kinases; Tyr153, Tyr201, and Tyr285, phosphorylated by JAK2; Tyr223, phosphorylated by CK2; and Thr212, phosphorylated by p35/Cdk5, Cdc2, and Erk1/2; and (iii) sites phosphorylated by both PAK1 and external kinases, such as Ser21, which is phosphorylated by AKT, PKG, PAK1, and Thr423, which is phosphorylated by PDK1 and PAK1 [31,32,33,34,35,36]. The activation state of PAK1 is further regulated by protein phosphatases, such as PP2A, which counteract phosphorylation events and fine-tune its activity [37]. These regulatory mechanisms provide the spatiotemporal control of PAK1 signaling, allowing for the precise modulation of cellular processes (Figure 1A).

PAK1 is susceptible to inhibition by a number of ATP-competitive and allosteric inhibitors that have been developed to counteract its dysregulated activity in diseases such as cancer [38,39,40]. ATP-competitive inhibitors occupy the ATP-binding pocket within the kinase domain, thereby preventing ATP access and blocking its catalytic activity. However, the development of highly selective inhibitors for PAK1 remains challenging due to the flexibility of its ATP-binding cleft and the significant sequence similarity it shares with other PAK family members [38]. Despite these challenges, several ATP-competitive inhibitors have been developed to target PAK1 activity, including G-5555, FL172, PF-3758309, AZ13705339, FRAX compounds, bis-anilinopyrimidine-based inhibitors, the aminopyrazole derivative compound 23, and ZMF-10 [38,39,40]. Allosteric inhibitors, on the other hand, bind to regulatory regions outside the ATP-binding pocket and induce conformational changes that interfere with PAK1 activation. These inhibitors often achieve greater specificity within the kinome by targeting regions that are less conserved among kinases. However, their potency is generally lower than that of ATP-competitive inhibitors because these binding sites are shallower and contain fewer key residues that contribute to strong inhibitor interactions [38]. Among the known allosteric inhibitors, IPA-3, NVS-PAK1-1, glaucarubinone, and the dibenzodiazepine compound 3 prevent PAK1 activation by interfering with its autoregulatory mechanism and upstream interactions [38,39,40]. The development of these inhibitors highlights the therapeutic potential of targeting PAK1 in diseases where its dysregulation plays a key role [40]. The classification of PAK1 inhibitors and their binding sites on PAK1 are shown in Figure 1A.

## 3. PAK1 Substrates

PAK1 exerts its activity primarily by phosphorylating a broad spectrum of substrates typically involved in maintaining cellular homeostasis. These substrates regulate processes such as energy production and consumption, the maintenance of genetic integrity, gene expression, the organization of cellular structures, movement, growth, programmed cell death, and differentiation (Figure 1) [4,41].

Energy metabolism: PAK1 phosphorylates and enhances the enzymatic activity of phosphoglucomutase 1 (PGM1), an important regulatory enzyme in cellular glucose utilization and energy homeostasis [42]. PGM1 catalyzes the reversible conversion between glucose-1-phosphate and glucose-6-phosphate. The actin polymerizing ARP2/3 subunit p41ARC is another PAK1 substrate in skeletal muscle cells [43]. It has been suggested that p41ARC links NWASP-cortactin-mediated actin polymerization and GLUT4 translocation to the plasma membrane by vesicle exocytosis.

DNA damage response: Microrchidia CW-type zinc finger 2 (MORC2) associates with chromatin and is phosphorylated by PAK1 at serine 739, promoting the PAK1 phosphorylation-dependent induction of gamma-H2AX, a critical step in the cellular response to DNA damage [44]. PAK1 is activated in response to UV-B radiation and then translocates to the nucleus to bind C-FOS, which acts as a transcriptional regulator of the ataxia-telangiectasia and RAD3-related protein (ATR) gene [45]. It has been reported that TCL/RHOJ-mediated activation of PAK1 in response to drug-induced DNA damage can suppress ATR [46].

Transcription: Nuclear PAK1 binds to the promoter region of the DNA repair kinase ATR through C-Fos, thereby regulating its transcription [45]. Nuclear PAK1 has also been shown to drive the transcription of fibronectin, which is critical for promoting the growth and migration of pancreatic cancer cells [47]. PAK1 phosphorylates FOXO1 to prevent its nuclear translocation and transcriptional activity [48]. PAK1-mediated phosphorylation stabilizes β-catenin and promotes TCF/LEF-dependent transcriptional activity [49,50].

Cell cycle progression: PAK1 contributes significantly to cell cycle progression by phosphorylating several proteins, including Aurora A, ARPC1b, PLK1, histone H3, MORC2, NFκB, and CRAF [40]. Increased expression of PAK1 in breast cancer cells increases cyclin D1 mRNA and protein levels and its nuclear accumulation [51]. Activated PAK1 phosphorylates and activates estrogen receptor-α (ERα), upregulates cyclin D1 expression through the NFκB pathway, and controls cell cycle progression [51]. In addition, CDK4 forms a complex with PAK1 in response to monocyte chemotactic protein 1 [52], activating the cyclin D1-CDK6-CDK4-PAK1 axis, which controls smooth muscle cell migration and proliferation. Silencing of PAK1 has been shown to reduce cyclin E and CDK2 expression, leading to cell cycle arrest at the G1 and S phases in hepatocellular carcinoma [53]. PAK1 also appears to be involved in the phosphorylation of retinoblastoma (Rb) and the activation of E2F transcription factors [54].

Cytoskeletal remodeling: PAK1 phosphorylates several key factors involved in cytoskeletal remodeling, including LIM kinase, p41–ARC, Filamin A, Paxillin, snail, and tubulin cofactor B (TCoB), thereby controlling various cellular processes. LIM kinase phosphorylates cofilin to prevent actin depolymerization [55,56]. The p41–ARC subunit of the Arp2/3 complex promotes actin nucleation and cell motility [57,58]. Filamin A, a filamentous actin cross-linking protein, anchors various cytoskeletal proteins and regulates cell adhesion and migration [59,60,61,62]. Paxillin phosphorylation triggers the formation of adhesion junctions and focal adhesions, thereby controlling cell contractility and motility [63,64,65]. PAK1 negatively modulates myosin light-chain phosphorylation by inhibiting myosin phosphatase target subunit 1 (MYPT1), increasing muscle contraction [66,67,68]. Snail phosphorylation by PAK1 controls epithelial cell permeability [69] and contributes to epithelial–mesenchymal transition (EMT) [70]. Phosphorylated by cyclin B1/CDK2, PAK1 controls microtubule reorganization by activating TCoB, a protein responsible for tubulin heterodimer assembly [71] and by inactivating stathmin, a microtubule destabilizing factor [72,73,74]. PAK1 regulates microtubule organization and spindle positioning during the metaphase-to-anaphase transition in conjunction with its activity against Aurora A, TACC3, and LIMK1 [66].

Cell proliferation and survival: PAK1 promotes cross-talk with RAF and PI3K signaling networks and contributes to cell proliferation and survival [41]. It phosphorylates CRAF and MEK1 and enhances ERK activity [75,76,77,78]. As a scaffolding protein, PAK1 modulates the PI3K-PDK1-AKT-mTOR pathway by facilitating the recruitment of AKT to the plasma membrane via PDK1, leading to AKT activation [79].

Apoptosis: PAK1 can interfere with the interaction of the BCL-2-associated agonist of cell death (BAD) with BCL2 and the induction of apoptosis in two ways [80]. PAK1 phosphorylates BAD directly [81] or indirectly through the phosphorylation of CRAF and its translocation to the mitochondria [80]. The shorter isoform of the PITSLRE protein kinase family, p110C, has been shown to bind to PAK1 and prevent BAD phosphorylation [82].

## 4. Accessory Proteins for PAK1

Accessory proteins are modulatory proteins that facilitate the spatiotemporal organization of signaling complexes during processing but are not directly involved in signal transduction [83]. They possess a variety of protein–protein interaction domains, motifs, and inherently disordered regions (IDRs) that allow them to interact with and link different signaling molecules. Thus, accessory proteins regulate and enhance signaling by directing the local assembly of protein complexes through multiple interactions. This is achieved either by reducing the dimensionality of the interactions or by increasing the local levels of interacting proteins [84]. Scaffolding, adaptor, anchoring, and docking proteins are the four major groups that can be categorized according to their structure and how they perform their functions [85].

In addition, accessory proteins can control interference between different signaling pathways, increase the residence time of proteins on the membrane, stimulate nanocluster formation, sequester effectors to inhibit their activation, and determine cell type specificity and subcellular localization of signaling cassettes. Because of their modulatory properties, accessory proteins can play critical roles in various situations. The expression of accessory proteins is tightly regulated but is often aberrant in malignant tumors (see below). Targeting accessory modulators rather than the core components of disease-relevant pathways has several advantages. Most importantly, this approach reduces abnormal signaling to physiological levels through strong suppression without eliminating them. This shifts the focus from the “inhibition” to “attenuation” of signaling [86].

The regulation of the subcellular localization of PAK1 is important in controlling its functions. For example, various stimuli localize and activate PAK1 at focal adhesion complexes and leading-edge membrane ruffles [87,88]. Below, we highlight the accessory proteins involved in PAK1 modulation and their roles in various types of cancers and developmental disorders, opening new avenues for therapeutic intervention, with these candidates emerging as promising targets. Based on their roles, accessory proteins for PAK1 can be classified into two groups, comprising modulators of PAK1 activation and modulators of PAK1 inhibition (Figure 1).

### 4.1. Modulators of PAK1 Activation

Many signaling molecules, including αPIX, βPIX, CKIP1, GIT1, GIT2, GRB2, NCK1, Paxillin, RIT1, and RIT2, positively modulate PAK1 activity (Figure 2).

α/βPIX: αPIX (also called COOL2 or ARHGEF6; 87.5 kDa; 776 amino acids) and βPIX (also called COOL1 or ARHGEF7; 90.0 kDa; 803 amino acids) are members of the DBL family RHOGEFs [19]. Each consists of a calponin homology (CH), an SH3 domain, a DBL homology (DH), a pleckstrin homology (PH), a proline-rich domain (PRD), a CAT/GIT binding domain (CBD), and a coiled-coil (CC) region. α/βPIX are generally accepted as GEFs for RAC1 and CDC42. Extensive biochemical analysis revealed that some DBL family proteins, including α/βPIX DH-PH tandems, lack GEF activity, suggesting that these proteins act as scaffolding proteins [89]. Accordingly, αPIX and βPIX interact with several proteins, including PAK1 [90], PAK4 [91], GIT1 [92], integrin-linked kinase (ILK), focal adhesion-associated β-Parvin [93], dynamin-2 [94], and BIN2/BRAP1 [95]. They are involved in integrin-dependent neurite outgrowth, the regulation of podosome size and number in macrophages, and podosome formation, motility, and phagocytosis in leukocytes. The loss of αPIX causes X-chromosomal nonspecific cognitive impairment [96].

The high-affinity interaction between the proline-rich region (amino acids 186–203) of PAK1 and the SH3 domain of αPIX controls PAK1 recruitment to focal complexes [90,97]. TCR-induced PAK1 activation in Jurkat cells depends on PAK1 complexing with PIX and GIT2 [98]. Accordingly, PAK1 can control cytoskeletal reorganization and transcriptional regulation in T cells. One key regulator of this process is CBL-B, an E3 ubiquitin ligase that competitively blocks the receptor-mediated PIX-PAK interaction [99]. In addition, CBL-B catalyzes the receptor ubiquitination, thereby exerting a dual regulatory effect, interfering with βPIX-modulated PAK1 activation and promoting receptor ubiquitination to modulate receptor signaling [99]. In addition, αPIX has been shown to modulate EGFR trafficking by controlling receptor recycling and degradation [100].

CKIP1: CKIP1 (also called PLEKHO1 or OC120; 46.2 kDa; 409 amino acids) is a casein kinase-2 α-subunit (CK2α)-interacting protein, consisting of an N-terminal PH domain and a C-terminal autoinhibitory leucine zipper with basic and acidic motifs and five proline-rich motifs distributed throughout the protein [101]. Due to its scaffolding properties, CKIP1 rapidly translocates to the plasma membrane in response to various signals, such as insulin, by binding to PIP3 [102]. This process recruits several signaling molecules, including CK2 [103], actin capping proteins [104], AP1 [105], IFP35 [106], ATM [107], and AKT [108], to modulate their activities. During apoptosis, CKIP1 is cleaved by caspase-3 and translocates to the nucleus, where its C-terminal cleavage product represses AP-1 activity [105]. Thus, CKIP1 controls several cellular processes, including morphogenesis, migration, and differentiation [109].

The ternary complex consisting of CK2α, CKIP-1, and PAK1, translocates to membrane ruffles in response to EGF, where CKIP-1 mediates the interaction between CK2α and PAK1 in a PI3K-dependent manner 1 [110]. This process results in the CK2-mediated phosphorylation of PAK1 at Ser223, leading to full PAK1 activation, as observed in prostate cancer cells. The inhibition of CKIP1 prevents PAK1-mediated actin cytoskeleton dynamics and cell migration without altering PAK1 expression and other PAK1 activities in these cells [110].

GIT1/2: GIT1 (also called CAT1, APP1; 84.3 kDa; 761 amino acids) and GIT2 (also called CAT2, or p95APP2, or PKL; 84.5 kDa; 759 amino acids) belong to the ARFGAP protein family and consist of an N-terminal ARFGAP domain, ankyrin repeats, SPA homology domain (SHD), synaptic localization domain (SLD), and Paxillin binding site (PBS).

GIT1 is detected in the vasculature and bronchial epithelia of the lung and in the vasculature and bile ducts of the liver, whereas GIT2 is ubiquitously expressed [111]. GIT1/2 regulates various cellular processes, including membrane trafficking, focal adhesion assembly and disassembly, migration, and chemotaxis, through the spatial localization of various signaling molecules, including GRKs, PLCγ, NOTCH1, mTOR, MEK1, ERK1, FAK, Paxillin, and α/βPIX via the activation of integrins, RTKs, and GPCRs [84,92,111,112,113,114,115]. GIT2 has also been reported to function as an integrator protein in aging processes [116]; GIT2 knockout mice age faster than age-matched WT controls [117].

GIT1/2 interact with α/βPIX to recruit PAK1, linking PAK1 activity to multiple signaling pathways in different cell types [118,119,120]. Increasing the local concentration of PAK1 can stimulate its kinase activity [121]. Recruiting PAK1 to the membrane at the T cell–APC junction increases its local concentrations, thereby promoting its activation [122]. GIT1/2 and α/βPIX form large complexes inside the cell [112,113], and this oligomerization is essential for adhesion site localization. Mutations that disrupt the GIT-PIX interaction or PIX homodimerization cause both proteins to disperse throughout the cytoplasm [121,123]. Thus, GIT1/2 promote the integration of multiple signaling pathways and regulate cell migration and protrusion.

GRB2: GRB2 (25.2 kDa; 217 amino acids) is a ubiquitously expressed adaptor protein consisting of three domains, namely a central SH2 domain flanked by two terminal SH3 domains. GRB2 binds to various signaling molecules and modulatory proteins and plays both positive and negative roles in controlling transmembrane receptor signaling [124]. Its best-studied function is to recruit the RAS activator SOS1 and link it to growth factor receptors [125,126].

GRB2 binds to PAK1 in response to EGF receptor activation via its second proline-rich motif (amino acids 40–45), an interaction that controls actin remodeling [127]. The transcriptional activity of the ERK1/2-KLF2 axis fine-tunes GRB2-modulated PAK1 activation, thereby influencing endothelial cell proliferation, migration, and angiogenesis [128]. The molecular nature of the GRB2-PAK1 complex formation in a cellular context is still unclear. A complex consisting of PAK1, SOS1, and KRAS has been detected in sensitized bone marrow mast cells [129]. Whether GRB2 is part of this complex is uncertain. However, a stoichiometric complex between GRB2, PAK1, and SOS1 seems unlikely because both SH3 domains of GRB2 are required to recruit and activate SOS1 at the plasma membrane [125]. Thrombin has been shown to trigger PAK1 activation in vascular smooth muscle cells via an EGFR-GAB1-SHP2-RAC1/CDC42-dependent mechanism [130]. GRB2 may be an important adaptor protein in this context, as it is directly associated with activated EGFR and GAB1 [84]. GAB1 acts as a docking platform for several signaling molecules, including SHP2, PLCγ, and PI3K, and it cross-links different signaling pathways [84].

NCK1: NCK1 (42.9 kDa; 377 amino acids) is a ubiquitously expressed adaptor protein consisting of three tandem SRC homology 3 (SH3) domains followed by a C-terminal SRC homology 2 (SH2) domain. The SH2 domain binds to specific phosphotyrosine residues of numerous transmembrane proteins, including growth factor receptors, ephrin receptors, cell adhesion molecules, T cell antigen receptors, and nephrin. NCK1 links extracellular signals to intracellular events by selectively targeting a large number of proline-rich region-containing signaling molecules with its three SH3 domains, thereby driving pathways related to cytoskeletal reorganization [131,132,133].

The direct association of the second SH3 domain of NCK1 with the very N-terminal proline-rich region (amino acids 12–18) of PAK1 is critical for the recruitment of PAK1 to activated transmembrane proteins at the plasma membrane [134,135]. NCK1 links various upstream signals to PAK1 and thereby determines the specificity of PAK1-controlled biological processes in a cell type-dependent manner. These processes include the promotion of endothelial paracellular pore formation and permeability [136], blastomere adhesion regulated by EPHA4 receptor tyrosine kinase [137], inflammation induced by oxidative stress [138], cell migration stimulated by AKT [139], and focal adhesion formation and cell migration induced by VEGF [140]. Interestingly, NCK1 is a substrate of EPHA4, which phosphorylates conserved tyrosines in the SH3 domains, thereby blocking SH3 interactions as a negative feedback mechanism [141].

Diseases associated with the NCK1-PAK1 interaction are diverse. NCK1 promotes angiogenesis in cervical squamous cell carcinoma and ovarian cancer via the RAC1/PAK1/MMP2 axis [142,143] and in colorectal cancer via the NCK1/STAT3/PAK1/ERK axis [144]. Similarly, the endothelial PDGF-B/NCK1/Pak1 axis promotes angiogenic sprouting and pathological neovascular tufting in ischemic retinopathy [145]. In contrast, NCK1 can interfere with the prolactin-activated JAK2/PAK1 axis, which plays a role in breast cancer promotion [146]. Targeting the NCK-PAK1 interaction could block the angiogenesis and neovascularization of tumor cells, providing a potential new approach to prevent ocular neovascular disease and improve retinal wound healing [145]. The small-molecule inhibitor AX-024 has been reported to bind to the SH3 domains of NCK1, interfere with its binding to the T cell receptor (TCR) subunit CD3ε, and block T cell activation [147,148]

Paxillin: The scaffold protein Paxillin (64.5 kDa; 591 amino acids) plays a key role in the assembly of focal adhesions and thus in cell migration [149]. It consists of an N-terminal region with five leucine–aspartate-rich (LD) motifs and a C-terminal region with four tandem LIN11-ISL1-MEC-3 (LIM) domains. These domains allow Paxillin to interact with a variety of proteins, including CRK, CSK, FAK, FRNK, GIT1, Merlin, p130Cas, PYK2, Talin, Vinculin, and SRC.

The association of GIT1, an ARFGAP-like protein, with Paxillin recruits PIX and PAK1 to the focal complex [115]. PAK1 activity has been shown to couple focal adhesion (FA) dynamics to leading-edge actin dynamics [150,151]. However, local regulation is not only triggered solely by mechanosensitive proteins. Signaling pathways, often triggered by protein phosphorylation, can occur at adhesions independently of tension [152,153]. For example, the PAK1-mediated phosphorylation of Paxillin at Ser-273 enhances adhesion dynamics and promotes the formation of the trimeric protein complex GIT1/PIX/PAK1, which associates with Paxillin after its phosphorylation [64,118].

RIT1/2: RIT1 (25.1 kDa; 219 amino acids) and RIT2 (24.7 kDa; 217 amino acids) belong to the RAS family of small GTPases, consisting of a central GDP/GTP-binding (G) domain flanked by variable amino acid extensions. They act as molecular switches in the cell, cycling between a GTP-bound active form and a GDP-bound inactive form [154,155]. RIT1/2 are classified as RAS-like without CAAX (C is cysteine, A is any aliphatic amino acid, and X is any amino acid), meaning they are not post-translationally isoprenylated like classical RAS proteins [156]. The ubiquitously expressed RIT1 and its neuron-specific paralog RIT2 were identified in 1996 as a novel subfamily of RAS-related proteins in the mouse retina [157]. Several studies using transgenic and knockout animal models have elucidated the emerging roles of RIT1/2 in regulating cell survival, proliferation, differentiation, and morphogenesis [155]. Genetic studies have implicated RIT1 and RIT2 signaling in both malignancies and developmental disorders, including Parkinson’s disease, autism, schizophrenia, and Noonan syndrome [158,159,160,161,162].

RIT1 has been reported to control actin dynamics and cell motility by forming a complex with PAK1, which also contains RAC1 or CDC42 [65]. The CRIB domain of PAK1 interacts directly with RIT1 in a nucleotide-independent manner. The ectopic expression of RIT1 leads to the disassembly of stress fiber and focal adhesions. The coexpression of dominant negative CDC42 or RAC1 and kinase-dead PAK1 prevents this effect [65].

### 4.2. Modulators of PAK1 Inhibition

Various proteins, including CRIPAK, Merlin, Nischarin, PAK1IP, and SKIP, negatively modulate PAK1 activity (Figure 2).

CRIPAK: CRIPAK (49 kDa; 446 amino acids) consists of an N-terminal C4 zinc finger domain followed by 12 C3H zinc finger domains. CRIPAK binds to ERα in a ligand-sensitive manner and also to PAK1 [163].

The binding site of CRIPAK in PAK1 has been localized to amino acids 132–270, which partially overlap with the Pak1 inhibitory domain (amino acids 83–149). The binding of CRIPAK may enhance the function of the autoinhibitory domain, thereby reducing PAK1 activation. Alternatively, because the CRIPAK’s binding site overlaps with the PIX binding region in Pak1, CRIPAK-mediated inhibition could result from its potential interference with the binding of other regulatory proteins, such as PIX [98,164].

The literature on the negative regulation of PAK1 by cysteine-rich inhibitors of PAK1 (CRIPAK) is limited. Talukder et al. [163] were the first to report CRIPAK as a PAK inhibitor, showing that it is widely expressed in various human cells and tissues. They identified CRIPAK as a novel PAK1-interacting protein. In addition, their data showed that CRIPAK binds to PAK1 through its N-terminal regulatory domain and blocks PAK1 kinase activity both in vitro and in vivo. CRIPAK prevents PAK1-mediated LIMK activation and contributes to estrogen receptor (ER) transactivation in breast cancer cells. On the other hand, the selective inhibition of endogenous CRIPAK resulted in increased PAK1 activity, thereby promoting cytoskeletal remodeling. Hormonal induction increased CRIPAK expression and enhanced its colocalization with the ER in the nucleus.

CRIPAK exerts a potent inhibitory effect on PAK1 activity through multiple mechanisms. First, the CRIPAK interaction region in PAK1 overlaps with the PAK1 autoinhibitory domain, implying that the CRIPAK-PAK1 interaction may enhance the function of the autoinhibitory domain, thereby reducing PAK1 activation. Second, since the CRIPAK binding site also overlaps with the PIX interaction region in PAK1, this interference may strongly disrupt the interaction of other regulatory partners. Physiologically, the loss of PAK1 inhibitors could lead to dysregulated PAK1 activation. Interestingly, the CRIPAK gene is located on chromosome 4p16.3, a region that is frequently deleted in breast tumors. This suggests that loss of CRIPAK in breast cancer cells could lead to persistent PAK1 activation, potentially contributing to breast carcinogenesis. Conversely, their data indicated that CRIPAK inhibits PAK1-mediated ER transactivation. In addition, 17-β-estradiol stimulates CRIPAK expression, while ER signaling enhances the nuclear localization of CRIPAK, where CRIPAK and ER colocalize in the presence of estrogen. These findings suggest that the loss of CRIPAK in breast tumors may enhance hormone independence via PAK1 regulation.

Merlin: Merlin (also called neurofibromin-2 or schwannomin; 69.7 kDa; 595 amino acids) is a member of the ezrin, radixin, and moesin (ERM) protein family, which is involved in signal transduction pathways by linking membrane proteins to the actin cytoskeleton. It consists of an N-terminal FERM domain, followed by an α-helical domain and a C-terminal domain. The latter physically binds to the FERM domain, forming a head-to-tail autoinhibited conformation that is released upon interaction with angiomotin [165]. Merlin binds phosphoinositides, including PIP2 [166], and binds PIKE-L, a brain-specific nuclear GTPase, to suppress PI3K activity [166]. Merlin also binds directly to ERBB2 [167], βII-spectrin [168], LATS1/2 [165], the VEGFR2–VE-cadherin complex at cell–cell junctions [169], the E3 ubiquitin ligase DCAF1 [170], the neurofibromin/Spred1 complex [171], the RAS/p120RASGAP complex [172], and β-catenin [173]. Consequently, the loss of Merlin activates RAC1 and RAS, leading to the hyperactivation of the PAK1, mTORC1, EGFR-RAS-ERK, PI3K-AKT, WNT, and LATS1/2 pathways [174,175].

Merlin binds directly to the PBD of PAK1, interfering with the binding of RAC1 and Paxillin. It also inhibits the recruitment of PAK1 to the focal adhesion complexes, thereby preventing its activation [176]. In addition, Merlin interacts with the tight junction protein angiomotin, which inhibits RAC1 activity [170]. Accordingly, Merlin deficiency in neurofibromatosis type 2 patients is associated with increased levels of GTP-bound RAC1 [177] and the hyperactivation of PAK1 [178]. Interestingly, constitutively active variants of RAC1 and PAK1 prevent Merlin from inhibiting RAS-induced activation of the MAPK pathway [179].

Nischarin: Nischarin (IRAS; 166.6 kDa; 1504 amino acids) consists of an N-terminal phosphoinositide-binding PHOX homology (PX) domain followed by leucine-rich repeats, a coiled-coil glutamine-rich region, an integrin-binding domain, and an alanine–proline-rich region. Nischarin interacts with several transmembrane proteins, including the integrin alpha5 subunit [180], the opioid receptor [181], the glutamate transporter GLT-1 [182], the adaptor protein IRS4 [183], the small GTPases RAB 4, 9, 14, and 38, and RAC1 [184], and kinases such as STK11/LKB1 [185], LIMK1 [186], and PAK1 [187,188]. It modulates various processes, including transcription, cytoskeletal reorganization, proliferation, survival, and differentiation [189,190,191]. Nischarin is expressed in various cell types and organs [189] and acts as a tumor suppressor gene when upregulated in cancer cells [192], inhibiting EMT and migration [191,193]. It is also associated with cardiac remodeling and dysfunction [194].

The upregulation of Nischarin in breast and colon epithelial cell lines has been shown to selectively inhibit RAC1/PAK1-induced migration and invasion [195]. Interestingly, Nischarin binds directly to the kinase domain of RAC1-activated PAK1 at the leading edge of a migrating cell, thereby inhibiting PAK kinase activity [187]. Endogenous Nischarin has also been shown to inhibit neurite outgrowth by blocking PAK1 activity at the neuronal membrane [196]. Nischarin’s inhibition of PAK1 prevents downstream effects such as the PAK1-mediated phosphorylation of LIMK, which would normally inhibit cofilin, an actin-severing protein, leading to actin filament assembly. By binding to the kinase domain of active LIMK, Nischarin deactivates it and prevents actin assembly during cytoskeletal reorganization [186].

PAK1IP1: PAK1IP1 (also known as hPIP1 or WDR84; 44 kDa; 392 amino acids) binds to the N-terminus of PAK1 and inhibits its activation by CDC42 [197]. PAK1IP1 is a human Gβ-like WD repeat protein, and its overexpression inhibits PAK-mediated c-JNK and NFκB signaling pathways [197]. Homozygous variants of PAK1IP1 have been shown to cause severe developmental defects of the brain and craniofacial skeleton, including a median orofacial cleft [198]. This may be related to the nuclear role of PAK1IP1 in ribosome biogenesis [199] or its interaction with MDM2 to upregulate p53 [198,200]. It remains unclear whether PAK1 is involved in this context. In addition, PAK1 translocates to the nucleus, where it may play a role in DNA repair and transcription [45,47].

SKIP: SKIP (also called INPP5K; 51.1 kDa; 448 amino acids) is a member of the phosphoinositide 5-phosphatase family. It consists of a 5-phosphatase domain followed by a C-terminal SKICH (SKIP carboxy homology) domain, essential for protein–protein interaction and subcellular localization [201]. SKIP binds several proteins, including MAD2L1BP, SODD, GRP78, and PAK1, and is involved in cell cycle control, adhesion, and migration processes.

SKIP’s interaction with glucose-regulated protein 78 (GPR78), an endoplasmic reticulum (ER) chaperone involved in the ER stress response and the unfolded protein response (UPR), is required for its localization to the ER. Ijuin and coworkers proposed a model in which cytosolic SKIP interacts directly with GRP78 via its SKICH domain under basal unstimulated conditions [202,203,204,205,206]. However, this model raises the unresolved question of how cytosolic SKIP interacts with GRP78, which is located on the luminal side of the ER [206].

Upon insulin stimulation, the SKIP-GRP78 complex translocates from the ER to the plasma membrane, where activated PAK1 displaces GRP78 and binds to SKIP via an 11-amino acid region within the kinase domain of PAK. This interaction links SKIP to the insulin receptor complex, resulting in PtdIns(3,4,5)P3 dephosphorylation, decreased AKT2 phosphorylation, and the subsequent inactivation of insulin signaling. Co-immunoprecipitation experiments confirmed the association of SKIP with GRP78 [207].

Reduced SKIP expression in cells or in SKIP-modified (PpsBrdm1/+) heterozygous mice increases insulin signaling sensitivity in muscle cells and reduces diet-induced obesity in these mice [208,209,210]. In addition, SKIP expression is increased in skeletal muscle tissue from high-fat-fed and diabetic/obese mice compared to wild-type mice, linking SKIP to these metabolic conditions [203]. Supplementation of the culture medium with a synthetic peptide that matches the sequence of the PAK1 kinase domain that interacts with SKIP increases insulin signaling in muscle cell lines, likely by disrupting the formation of the SKIP-PAK1 complex [204].

As a mechanistic insight, Ijuin et al. [211] identified how SKIP-PAK1 interactions affect insulin signaling and glucose uptake. Upon insulin stimulation, SKIP translocates to membrane ruffles and binds activated PAK1 from a complex with PIP3 effectors such as AKT2, PDK1, and Rac1, leading to the inactivation of certain signaling proteins. SKIP can inhibit both the Rac1-dependent kinase activity and the scaffolding functions of PAK1. This inhibition limits Rac1 activity, which subsequently inactivates PAK1 and dissociates the AKT2-PDK1-PAK1 complex, allowing for rapid termination of the insulin signaling cascade.

These findings suggest that targeting the SKIP-PAK1 interaction may be a promising therapeutic strategy to improve systemic insulin-dependent glucose uptake and treat hyperglycemia. Notably, recent studies have identified an 11-amino acid peptide within the kinase domain of PAK1 that is essential for its interaction with SKIP. Expression of this peptide sequence in skeletal muscle cells enhances insulin signaling, and supplementation with a synthetic peptide of this sequence can improve insulin signaling and glucose uptake in skeletal muscle cell lines [204].

### 4.3. PAK1 and Focal Adhesion Complex Formation

Focal adhesions are dynamic multi-protein assemblies that mediate interactions between the extracellular matrix and the actin cytoskeleton, playing a pivotal role in cell migration and mechanosensing [212]. PAK1 is tightly regulated within focal adhesions by its interactions with scaffolding proteins such as αPIX, GIT1, and Paxillin (Figure 2) [97,98,99,115,118,119,120]. These accessory proteins coordinate the spatial localization and activity of PAK1, ensuring precise signal transduction within the adhesion complex. A key mechanism for the spatial regulation of PAK1 involves its association with the PAK1-PIX-GIT complex. This interaction is stabilized by the SH3 domain of αPIX, which binds to the proline-rich region of PAK1. Phosphorylation at Y285, downstream of cytokine receptor activation (including JAK2-mediated pathways), further promotes the assembly of the PAK1-PIX-GIT complex [31]. Once recruited to focal adhesions, αPIX and GIT stabilize PAK1 at these sites, facilitating its interactions with other adhesion components. Paxillin, an adaptor protein, links the PAK1-PIX-GIT complex to core focal adhesion components, such as focal adhesion kinase (FAK), by directly binding to GIT1. This connection promotes the assembly and maturation of focal adhesions and integrates PAK1 into signaling pathways that regulate cytoskeletal remodeling and focal adhesion dynamics. The PAK1-mediated phosphorylation of Paxillin at S273 further enhances these processes, highlighting its role in modulating cell migration [31]. Negative regulators also play a critical role in modulating PAK1 activity at focal adhesions. CRIPAK binds to a site on PAK1 overlapping the PIX interaction region, disrupting interactions with other regulatory partners [163]. Nischarin negatively regulates PAK1 by directly binding to its kinase domain, inhibiting its activity and destabilizing the PAK1-PIX-GIT-Paxillin complex [187,195]. Merlin, another regulatory protein, binds to the PBD of PAK1, interfering with RAC1 and Paxillin binding [176]. This prevents PAK1 recruitment to focal adhesion complexes and inhibits its activation. Collectively, these regulatory interactions ensure that PAK1 activity is tightly controlled within focal adhesions, maintaining cellular homeostasis and preventing aberrant signaling (Figure 2) [213].

### 4.4. PAK1 Membrane Localization, Activation, and Function

The recruitment of PAK1 to the plasma membrane is a critical step for its activation and subsequent signaling [214]. Accessory proteins such as GIT1, PIX, NCK1, and GRB2 play essential roles in mediating this localization and activation, ensuring precise spatiotemporal regulation of PAK1 activity. Upon receptor tyrosine kinase (RTK) activation by growth factors, RTKs undergo dimerization and phosphorylation, creating docking sites for adaptor proteins like NCK1 and GRB2 [127,128,215]. These adaptors recruit PAK1 to the plasma membrane, where it interacts with active GTP-bound RAC1 or CDC42 via its CRIB motif. In resting cells, RhoGDI sequesters RAC1 in the cytoplasm by interacting with its lipid anchor. Upon stimulation, RhoGEFs facilitate the exchange of GDP for GTP on RAC1, promoting its translocation to the plasma membrane. RhoGAPs deactivate RAC1 by accelerating GTP hydrolysis. Once localized to the membrane, PAK1 undergoes a conformational change upon binding to GTP-bound RAC1/CDC42, relieving its autoinhibition [19,65]. This leads to phosphorylation at Thr-423 within its kinase domain, a critical step for its activation. RIT1 stabilizes the interaction of PAK1 with RAC1 and CDC42, further enhancing its activation and downstream signaling [65]. Active PAK1 initiates diverse signaling pathways, including those involved in cytoskeletal remodeling, transcriptional regulation, cell cycle progression, survival, and migration. Several accessory proteins serve as negative regulators of PAK1 at the membrane [19]. CRIPAK exerts its inhibitory effects through multiple mechanisms: it overlaps with the PIX interaction region, disrupts regulatory interactions, and enhances PAK1 autoinhibition [163]. PAK1IP1 binds to the N-terminus of PAK1, blocking its activation. Nischarin inhibits PAK1 by binding directly to its kinase domain, suppressing its ability to phosphorylate substrates [187,195]. Merlin interferes with RAC1 binding by targeting the PBD of PAK1, thereby preventing its activation and membrane localization [176]. These finely tuned regulatory mechanisms highlight the importance of accessory proteins in controlling PAK1 membrane localization and activity, ensuring balanced signaling outcomes. Such a regulation is critical for maintaining cellular processes and preventing pathological conditions driven by aberrant PAK1 activation (Figure 2) [40,216].

## 5. PAK1 in Cancer Therapy and Resistance

Recent advances highlight the therapeutic potential of targeting PAK1 as a strategy to address cancer progression and therapy resistance [40]. In melanoma, PAK1 hyperactivation promotes cell survival, proliferation, and immune evasion, contributing to resistance against BRAF and MEK inhibitors. Studies demonstrate that PAK1 inhibition enhances the efficacy of these therapies by modulating the MAPK and PI3K/AKT pathways. A novel PAK1-selective degrader, BJG-05-039, combines targeted degradation with kinase inhibition, providing a dual mechanism to suppress both the enzymatic and scaffolding functions of PAK1. Synergistic approaches, such as combining PAK1 inhibitors with immune checkpoint therapies, are also being explored to boost anti-tumor immunity. While promising, challenges related to drug specificity, bioavailability, and toxicity remain key barriers to clinical translation [217].

In various cancers, PAK1 activation is closely linked to resistance mechanisms against standard therapies. For instance, in non-small cell lung cancer (NSCLC), PAK1 enhances β-catenin-mediated cancer stemness, stabilizing markers such as OCT4 and SOX2 and driving tumor aggressiveness and chemoresistance. Clinical evidence indicates that high PAK1 expression correlates with poor responses to cisplatin therapy. Combining MEK/ERK inhibitors like AZD6244 or targeting β-catenin-driven stemness with agents such as BBI-608 restores cisplatin sensitivity, offering a promising avenue for overcoming resistance. Similarly, in BRAF-mutant melanomas, PAK1 activation drives resistance by engaging the AKT signaling pathway. Targeting PAK1-regulated Wnt/β-catenin signaling has shown potential to improve drug sensitivity across multiple cancers [218].

In estrogen receptor-positive (ER+) breast cancer, PAK1 hyperactivation mediates resistance to endocrine therapy (ET) and CDK4/6 inhibitors by driving epithelial-to-mesenchymal transition (EMT) and MAPK pathway activation. Pharmacological inhibitors like PF-3758309 and NVS-PAK1-1 restore drug sensitivity by suppressing tumor invasion and growth. Combining PAK1 inhibitors with ET and CDK4/6 inhibitors represents a promising strategy to address resistance and improve patient outcomes in this aggressive breast cancer subtype [219].

The concept of modulating PAK1 rather than completely inhibiting it is emerging as a refined therapeutic approach to attenuate aberrant cancer signaling. By targeting accessory proteins involved in PAK1 regulation and modulation, it may be possible to fine-tune PAK1 activity, reducing cancer invasion and drug resistance without fully blocking essential cellular processes. This paradigm shift opens the door to innovative therapies that focus on balancing signaling dynamics, offering patients more effective and sustainable treatment options.

## 6. Conclusions and Perspective

Over the past three decades, novel pathway components, structural insights, biophysical principles, biomimetic strategies, and clinical drug candidates have emerged in the field of receptor-driven PAK1 signaling. As outlined, several accessory proteins, each with different sizes and domain architectures, regulate the binding and function of molecular components within the PAK1 pathways. These proteins orchestrate PAK1 assembly and activation in a context-dependent manner. The specificity, efficacy, and fidelity of signaling are largely determined by the spatial localization and temporal dynamics of these interactions, protecting against potential deleterious effects.

PAK1 serves as a key player in various pathological conditions, particularly cancer, making it a promising target for therapeutic agents or blockers. However, the direct inhibition of core signaling components, such as PAK1, often results in the complete shutdown of signaling. This blockade can lead to the emergence of multi-characteristics in cancer cells, including new features that allow them to evade therapy and develop drug resistance [219,220]. Such effects can contribute to increased off-target effects and enhanced aggressiveness of the tumor. Therefore, the traditional “inhibitor” concept, which focuses on completely blocking core pathway components, may need to be reassessed. Therapeutic strategies could benefit from exploring more refined approaches that modulate signaling, potentially minimizing these unintended consequences.

In this context, accessory proteins play a crucial role in modulating signaling pathways. These proteins ensure the strength, efficiency, and specificity of signal transduction by regulating the compartmentalization and site-specific localization of signaling molecules [84]. This regulation ensures that signaling events occur accurately and within the correct cellular context. Accessory proteins also offer dynamic control over core signaling components, such as PAK1, influencing critical cellular processes like migration, cytoskeletal remodeling, and survival. As such, they represent promising therapeutic targets because they allow for a more precise modulation of signaling without the need for the complete inhibition of core pathway components.

Among the various accessory proteins that modulate signaling pathways, several are key in regulating PAK1 activation and inhibition. For example, targeting PIX, a scaffolding protein that modulates PAK1 localization, and Paxillin, which plays a central role in the assembly of focal adhesions and cell migration, could potentially reduce PAK1 signaling and focal adhesion complex formation. Furthermore, targeting adaptor proteins like NCK and GRB2, as well as RIT1, may decrease PAK1 membrane localization and activation by RAC1/CDC42, thereby downregulating PAK1 signaling. Another promising approach is to explore strategies that stabilize the interaction of modulators of PAK1 inhibition, thereby extending the residence time required for PAK1 activation, which could provide a novel method for fine-tuning PAK1 signaling.

The inhibition of CKIP1 has been shown to prevent PAK1-driven actin cytoskeleton remodeling and cell migration without affecting PAK1 expression or other activities [110]. The small-molecule inhibitor AX-024 binds to the SH3 domains of NCK1, disrupting its interaction with the TCR subunit CD3ε and inhibiting T cell activation [148]. These findings highlight the potential of targeting accessory proteins to achieve a more targeted and controlled modulation of PAK1 activity. However, the discovery of inhibitors targeting accessory proteins remains limited, and this represents a major gap in the development of more effective therapies.

Alternative strategies to direct PAK1 inhibition, such as targeting accessory proteins, have been explored and show great promise for fine-tuning PAK1 signaling. For example, Chow et al. introduced a PAK1-selective degrader, which may offer more potent pharmacological effects than traditional catalytic inhibition [221]. Another approach focuses on targeting long noncoding RNAs (lncRNAs) associated with the PAK1 pathways. LncRNAs regulate gene expression through various mechanisms, including genomic imprinting, epigenetic regulation, and miRNA sponging, which could open up new therapeutic avenues [222]. Zhou et al. demonstrated that lncRNA-H19 activates the CDC42/PAK1 pathway to promote proliferation and invasion in hepatocellular carcinoma [223]. Similarly, Luo et al. showed that lncRNA MALAT1 facilitates BM-MSC differentiation into endothelial cells and improves erectile dysfunction via the miR-206/CDC42/PAK1/Paxillin axis [224]. These findings suggest that targeting lncRNAs in RAC1/CDC42 and PAK1 pathways could offer promising therapeutic outcomes.

Although the current literature is somewhat limited, existing data highlight the importance of a precisely controlled spatiotemporal organization of PAK1 by accessory proteins. Future research should explore these modulators to develop novel therapeutic strategies. Harnessing this level of control over PAK1 activity holds great promise for advancing disease treatment, offering a more refined approach to combatting diseases like cancer with limited undesirable side effects of direct pathway inhibition.

## Figures and Tables

**Figure 1 biomolecules-15-00242-f001:**
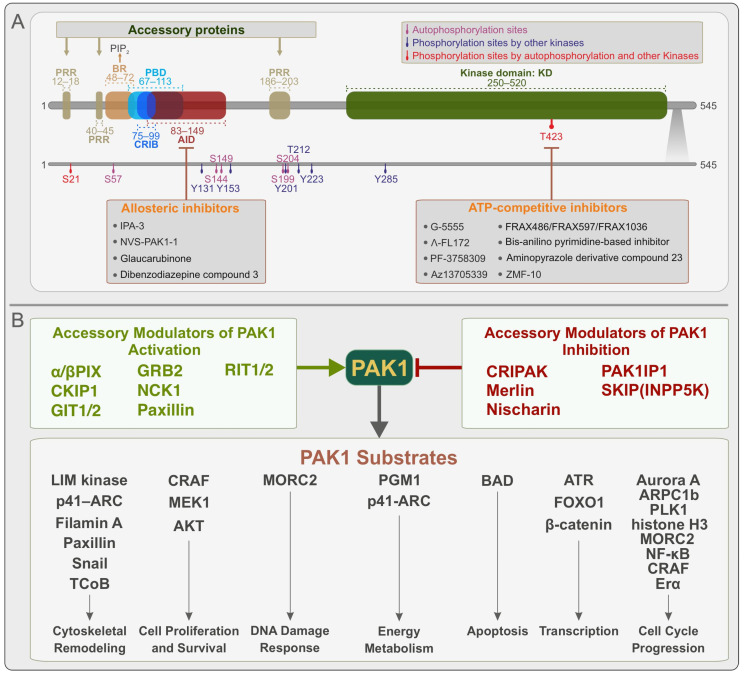
Overview of PAK1 domain organization, activation, inhibition, and interaction sites and key modulators and substrates. (**A**) Schematic representation of the domain organization of PAK1 (60.6 kDa; 545 amino acids), highlighting modulation and inhibition sites. See the text for further details. (**B**) Summary of accessory proteins that act as modulators of PAK1 activity and a diverse set of PAK1 substrates involved in maintaining cellular homeostasis.

**Figure 2 biomolecules-15-00242-f002:**
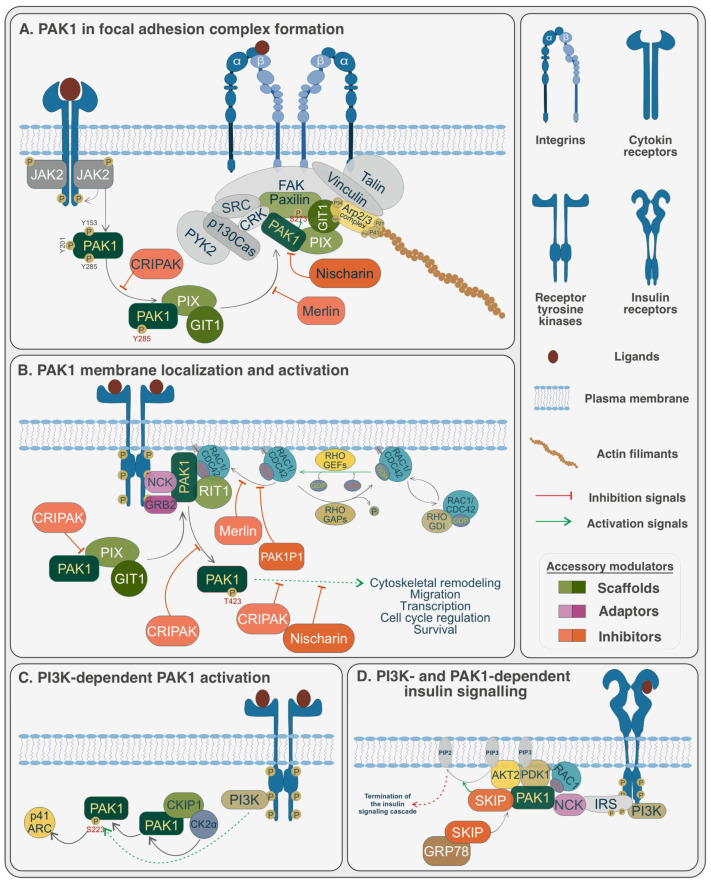
Schematic illustration of PAK1 modulation by accessory proteins. (**A**) PAK1 and focal adhesion complex formation: PAK1 interacts with scaffolding proteins such as αPIX and GIT and focal adhesion components like Paxillin to regulate adhesion assembly and cytoskeletal remodeling. Negative regulators, including Nischarin, CRIPAK, and Merlin, interfere with PAK1 binding, thereby inhibiting focal adhesion stability and PAK1 activity. (**B**) PAK1 membrane localization, activation, and function: PAK1 is recruited to the plasma membrane via adaptor proteins such as NCK and GRB2 and becomes activated through its interaction with GTP-bound RAC1. Once activated, PAK1 participates in various downstream pathways. Additional regulatory proteins, including CRIPAK, Nischarin, Merlin, and PAK1IP1, limit PAK1 activation or its kinase activity to maintain balanced signaling. (**C**) PI3K-dependent activation of PAK1: PAK1 is phosphorylated by the CK2α-CKIP-1 complex in a PI3K-dependent manner, which promotes its activity at membrane ruffles and enables the phosphorylation of substrates such as p41-Arc, driving cell signaling. (**D**) PI3K- and PAK1-dependent insulin signaling: Insulin receptor activation leads to IRS1 phosphorylation and PAK1 membrane recruitment, mediated by NCK. Active PAK1 interacts with PDK1 and AKT2 at the membrane, while the SKIP-GRP78 complex translocates from the ER to the membrane. SKIP binds PAK1, converting PIP3 to PIP2 and terminating the insulin signaling cascade.

## Data Availability

It is affirmed that no new data were generated while compiling this review manuscript. All referenced data sources are openly accessible and appropriately cited within the manuscript. Please do not hesitate to contact the corresponding author if any additional information or clarification is required.

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
