# Peer review of "Modulating PAK1: Accessory Proteins as Promising Therapeutic Targets"

_biomolecules, 2025, doi:10.3390/biom15020242_

Round 1

Reviewer 1 Report

Comments and Suggestions for Authors

In this manuscript, Mirzaiebadizi et al. reviewed the p21-activated kinase (PAK1) and its accessory proteins, focusing on aberrant activation of PAK1 in diseases and their potential as therapeutic targets especially that of accessory proteins. They outlined background of PAK1, reviewed its upstream regulators, downstream substrates, accessory proteins (activators and suppressors), and their roles in cancers. Finally, they proposed that the accessory proteins, not PAK1 itself, as promising therapeutic targets with some examples, based on the idea that inhibition of PAK1 itself may cause adverse effects and modification of accessory proteins may revert to physiological PAK1 activity.

The manuscript is well organized, covered adequate literature and is concisely wrapped up. The proposal that, not PAK1 itself, but its accessory proteins as therapeutic targets is intriguing. However, the manuscript could be improved by addressing following points.

Major point.

The manuscript is too concisely wrapped up and contains many abbreviated technical terms. Thus there seems to be some points, which may be difficult to understand for readers, who are not familiar with signal transduction. For examples, line 87-88 “GLUT4 vesicle translocation”. This may be difficult to understand without knowing that GLUT4 is stored in plasma membrane of intracellular vesicles and, upon stimulation with insulin, the vesicles fuses with cellular membrane, thereby increasing the amount of GLUT4 on cellular membrane. Adding this kind of background knowledge all through the text would facilitate understanding of the contents for readers, who are not familiar with signal transduction. It would also be kind for readers to add “glucose transporter” before “GLUT4”, and the same is true for other abbreviated technical terms without explanations.

Minor points.

1. Figure 1A. 423 of Thr423 is too small to see.

2. Lines 58-59. “Overview of PAK1 domain organization, inhibition and interaction sites, and key modulators and substrates.” No need for “activation”?

3. Lines 75-76. “PAK1 is activated by the CDC42 and RAC groups of small GTPases through several upstream signals, including estrogen, insulin, EGF, PDGF, IGF-1, and thrombin [32-34].” It may be better to move this sentence into “2. Mechanism of PAK1 Activation”.

4. Lines 93-94. “C-FOS, which acts as a transcriptional regulator of ataxia-telangiectasia and RAD3-related protein (ATR)”. It would be better to describe “C-FOS, which acts as a transcriptional regulator of ataxia-telangiectasia and RAD3-related (ATR) gene”, since it is the issue of transcription.

Author Response

In this manuscript, Mirzaiebadizi et al. reviewed the p21-activated kinase (PAK1) and its accessory proteins, focusing on aberrant activation of PAK1 in diseases and their potential as therapeutic targets especially that of accessory proteins. They outlined background of PAK1, reviewed its upstream regulators, downstream substrates, accessory proteins (activators and suppressors), and their roles in cancers. Finally, they proposed that the accessory proteins, not PAK1 itself, as promising therapeutic targets with some examples, based on the idea that inhibition of PAK1 itself may cause adverse effects and modification of accessory proteins may revert to physiological PAK1 activity.

The manuscript is well organized, covered adequate literature and is concisely wrapped up. The proposal that, not PAK1 itself, but its accessory proteins as therapeutic targets is intriguing. However, the manuscript could be improved by addressing following points.

Authors Response: We would like to thank the reviewer for taking the time and effort to read and review our manuscript. All changes are highlighted in yellow.

Major point:

The manuscript is too concisely wrapped up and contains many abbreviated technical terms. Thus there seems to be some points, which may be difficult to understand for readers, who are not familiar with signal transduction. For examples, line 87-88 “GLUT4 vesicle translocation”. This may be difficult to understand without knowing that GLUT4 is stored in plasma membrane of intracellular vesicles and, upon stimulation with insulin, the vesicles fuses with cellular membrane, thereby increasing the amount of GLUT4 on cellular membrane. Adding this kind of background knowledge all through the text would facilitate understanding of the contents for readers, who are not familiar with signal transduction. It would also be kind for readers to add “glucose transporter” before “GLUT4”, and the same is true for other abbreviated technical terms without explanations.

Authors Response: We appreciate the reviewer's feedback regarding the clarity of the abbreviated technical terms. To improve accessibility for readers less familiar with signal transduction, we have added a specific list of abbreviations at the end of the manuscript, which lists the abbreviations used throughout the text. This addition ensures that technical terms are clearly defined without disrupting the flow of the main text. In addition, we have gone through the text and completed text such as "GLUT4 vesicle translocation" to "GLUT4 translocation to the plasma membrane by vesicle exocytosis".

Minor points:

  1. Figure 1A. 423 of Thr423is too small to see.

Authors Response: The font size of all phosphorylation sites in Figure 1A has been increased for better visibility.

  1. Lines 58-59. “Overview of PAK1 domain organization, inhibition and interaction sites, and key modulators and substrates.” No need for “activation”?

Authors Response: Thank you for pointing this out. The figure legend title has been updated to "Overview of PAK1 domain organization, activation, inhibition and interaction sites, and key modulators and substrates."

  1. Lines 75-76. “PAK1 is activated by the CDC42 and RAC groups of small GTPases through several upstream signals, including estrogen, insulin, EGF, PDGF, IGF-1, and thrombin [32-34].” It may be better to move this sentence into “2. Mechanism of PAK1 Activation”.

Authors Response: We agree with the reviewer’s suggestion. The sentence has been moved to the section “2. Mechanism of PAK1 Activation” for better contextual placement.

  1. Lines 93-94. “C-FOS, which acts as a transcriptional regulator of ataxia-telangiectasia and RAD3-related protein (ATR)”. It would be better to describe “C-FOS, which acts as a transcriptional regulator of ataxia-telangiectasia and RAD3-related (ATR) gene”, since it is the issue of transcription.

Authors Response: We appreciate the reviewer’s suggestion to clarify the transcriptional role of C-FOS. The sentence has been revised for greater precision to reflect that C-FOS acts as a transcriptional regulator of the ataxia-telangiectasia and RAD3-related (ATR) gene.

Reviewer 2 Report

Comments and Suggestions for Authors

This review offers a clear summary of domain structure and activation mechanisms of PAK1, with a focus on the accessory proteins regulating its activity. This provides useful information for researchers in the field,and I would recommend its acceptance pending necessary revision.

Comments: 

Line 54: Change "the C-terminus" to "the C-terminal region" to match the earlier reference to " N-terminal, non-catalytic region" 

Line 55: The manuscript only mentions the phosphorylation site at Thr423, but there are other phosphorylation sites in the N-terminal, non-catalytic region of PAK1. These sites are critical for stabilizing the active conformation of PAK1 and involve multiple kinases. It is recommended to expand the section on accessory proteins to include kinases and phosphatases involved in PAK1 phosphorylation regulation. Additionally, Figure 1 should be revised accordingly.

Lines 75-76: These sentences are not directly related to this section. They would be better placed in the section on PAK1 activation mechanisms. 

Line 327: The sentence should be revised to: “RIT1/2 are classified as RAS-like GTPases without the CAAX motif.” 

Figure 1: Correct "dimain" to "domain." Several inhibitors are listed in the figure but are not clearly discussed in the manuscript. It would be helpful to mention them in more detail in line 72 within the context of PAK1 activity. Additionally, clarify whether "targets" and "substrates" of PAK1 refer to different entities in different contexts. 

Figure 2: The font size of the protein names is inconsistent. A uniform font size would enhance the figure's presentation.

Author Response

This review offers a clear summary of domain structure and activation mechanisms of PAK1, with a focus on the accessory proteins regulating its activity. This provides useful information for researchers in the field,and I would recommend its acceptance pending necessary revision.

Authors Response: We sincerely thank the reviewer for taking the time and effort to read and review our manuscript. All changes are highlighted in yellow.

Comments:

Line 54: Change "the C-terminus" to "the C-terminal region" to match the earlier reference to " N-terminal, non-catalytic region" 

Authors Response: We thank the reviewer for this suggestion. The term "the C-terminus" has been updated to "the C-terminal region" in line 54 (now line 56 in the revised version) to ensure consistency with the earlier reference to the "N-terminal, non-catalytic region."

Line 55: The manuscript only mentions the phosphorylation site at Thr423, but there are other phosphorylation sites in the N-terminal, non-catalytic region of PAK1. These sites are critical for stabilizing the active conformation of PAK1 and involve multiple kinases. It is recommended to expand the section on accessory proteins to include kinases and phosphatases involved in PAK1 phosphorylation regulation. Additionally, Figure 1 should be revised accordingly.

Authors Response: We appreciate the reviewer's valuable feedback. In response, we have expanded the manuscript to include additional phosphorylation sites within the N-terminal non-catalytic region of PAK1 that are essential for stabilizing its active conformation and regulating its signaling activity. These phosphorylation sites are now classified into three groups: (i) autophosphorylation sites, (ii) sites phosphorylated by other kinases, and (iii) sites phosphorylated by both PAK1 and external kinases. This expanded section has been included on line 74 to provide a more comprehensive overview of PAK1 phosphorylation regulation. In addition, Figure 1 has been revised to reflect these phosphorylation sites and categorized based on their regulatory mechanisms for clarity.

Lines 75-76: These sentences are not directly related to this section. They would be better placed in the section on PAK1 activation mechanisms. 

Authors Response: We appreciate the reviewer’s suggestion. The sentences in lines 75-76 have been relocated to the section titled “2. Mechanism of PAK1 Activation” to ensure better contextual alignment and improve the flow of the manuscript.

Line 327: The sentence should be revised to: “RIT1/2 are classified as RAS-like GTPases without the CAAX motif.” 

Authors Response: We thank the reviewer for the suggestion. The sentence has been revised to: “RIT1/2 are classified as RAS-like GTPases without the CAAX motif,” as recommended.

Figure 1: Correct "dimain" to "domain." Several inhibitors are listed in the figure but are not clearly discussed in the manuscript. It would be helpful to mention them in more detail in line 72 within the context of PAK1 activity. Additionally, clarify whether "targets" and "substrates" of PAK1 refer to different entities in different contexts. 

Authors Response: We appreciate the reviewer's valuable suggestions. The typographical error "dimain" has been corrected to "domain" in Figure 1. In response to the comment regarding inhibitors, we have added a dedicated paragraph in the manuscript discussing ATP-competitive and allosteric PAK1 inhibitors, their mechanisms of action, and examples to provide a clearer contextual link to PAK1 activity (now on line 89 of the revised version). In addition, to improve clarity and accuracy, we have removed the word "targets" from the phrase "PAK1 targets and substrates" in Figure 1, as PAK1 substrates are inherently its targets.

Figure 2: The font size of the protein names is inconsistent. A uniform font size would enhance the figure's presentation.

Authors Response: We appreciate the reviewer’s feedback. The font size of all protein names in Figure 2 has been standardized to ensure consistency and improve the overall presentation of the figure.